# A scoping review of patient-centred tuberculosis care interventions: Gaps and opportunities

**Hanlie Myburgh**[1,2]*, **Dzunisani Baloyi**[1], **Marian Loveday**[3,4], **Sue-Ann Meehan**[1], **Muhammad Osman**[1,5], **Dillon Wademan**[1], **Anneke Hesseling**[1], **Graeme Hoddinott**[1]

**1** Desmond Tutu TB Centre, Department of Paediatrics and Child Health, Faculty of Medicine and Health Sciences, Stellenbosch University, Cape Town, South Africa, **2** Amsterdam Institute for Social Science Research (AISSR), University of Amsterdam, Amsterdam, The Netherlands, **3** HIV and other Infectious Diseases Research Unit (HIDRU), South African Medical Research Council (SAMRC), Cape Town, South Africa, **4** Centre for Health Systems Research & Development, University of the Free State, Bloemfontein, South Africa, **5** School of Human Sciences, Faculty of Education, Health & Human Sciences, University of Greenwich, London, United Kingdom

* hmyburgh@sun.ac.za

**Data Availability Statement:** Data sharing is not applicable to this article as no datasets were generated or analysed during the current study.

## Abstract

Tuberculosis (TB) is a leading cause of death globally. In 2015, the World Health Organization hailed patient-centred care as the first of three pillars in the End TB strategy. Few examples of how to deliver patient-centred care in TB programmes exist in practice; TB control efforts have historically prioritised health systems structures and processes, with little consideration for the experiences of people affected by TB. We aimed to describe how patient-centred care interventions have been implemented for TB, highlighting gaps and opportunities. We conducted a scoping review of the published peer-reviewed research literature and grey literature on patient-centred TB care interventions between January 2005 and March 2020. We found limited information on implementing patient-centred care for TB programmes (13 research articles, 7 project reports, and 19 conference abstracts). Patient-centred TB care was implemented primarily as a means to improve adherence, reduce loss to follow-up, and improve treatment outcomes. Interventions focused on education and information for people affected by TB, and psychosocial, and socioeconomic support. Few patient-centred TB care interventions focused on screening, diagnosis, or treatment initiation. Patient-centred TB care has to go beyond programmatic improvements and requires recognition of the diverse needs of people affected by TB to provide holistic care in all aspects of TB prevention, care, and treatment.

## 1. Introduction

Tuberculosis (TB) is one of the leading causes of death globally due to a single infectious agent [1]. In 2020, an estimated 10 million people developed TB disease, and for the first time in three years the number of deaths has increased, with more than 1.5 million deaths estimated [1].

**Funding:** This publication is based on research funded by the Bill & Melinda Gates Foundation through their support of the South African National TB Think Tank. The findings and conclusions contained within are those of the authors and do not necessarily reflect positions or policies of the Bill & Melinda Gates Foundation.

**Competing interests:** The authors have declared that no competing interests exist.

Historically, TB control efforts have prioritised biomedical and public health approaches that focus on rapid passive detection, strict case notification, and surveillance–alongside close adherence monitoring through Directly Observed Therapy (DOT) [2, 3]. These strategies have been critiqued for being paternalistic in their approach to managing people affected by TB [4–6], for using language that is disempowering [7, 8], and for being indicative of relationships of mistrust between patients and providers [6, 9]. While conventional surveillance responses to the TB epidemic have shown some success as a TB control strategy, persistent shortfalls in the performance of National TB Programmes along the TB care cascade remain [10–12].

In 2015, the World Health Organization (WHO) hailed patient-centred care as the first of three pillars of its End TB strategy, considering it essential to achieve the End TB targets and TB-related Sustainable Development Goals [13]. Patient-centred care is broadly defined as care that fosters relationships of partnership and trust between patients and providers, and which is holistic, individualised, empowering, and respectful of patients' contexts, needs, and autonomy [14–18]. While there is growing recognition of the importance of patient-centred care for TB, translating patient-centred care into practice within TB programmes is complex because its principles are in tension with TB control strategies [15, 19]. However, Horter et al. suggest how patient-and person-centred care approaches can facilitate and support the public health goals of TB control programmes though few examples exist in practice [6].

We aimed to highlight gaps and opportunities for implementing patient-centred TB care by 1) identifying examples of interventions that implement patient-centred TB care, 2) considering the strengths, limitations, and lessons learnt from these interventions, and 3) proposing ways in which the principles of patient-centredness can be routinely implemented in TB programmes.

## 2. Methods

We conducted a scoping review of the published peer-reviewed research literature and grey literature. A scoping review methodology [20] is an appropriate method for this research as patient-centred care for TB is novel and limited. A scoping review methodology allowed us to iteratively and reflexively 'map' and explore patient-centred TB care interventions across multiple information sources, including research databases, the Google search engine, TB conference platforms, TB-related websites, reference lists, and through discussions with researchers and TB programme implementers. We followed the preferred reporting items for systematic reviews and meta-analyses extension for scoping reviews (PRISMA-ScR) [21] to report our scoping review process and findings (see S1 Appendix).

### 2.1 Review inclusion and exclusion criteria

The review aimed to identify examples of how patient-centred care was implemented or 'put into practice' in TB programmes. Many studies on patient-centred care highlight the ambiguities inherent to the concept, i.e., in some understandings any intervention that aims to improve patient outcomes can be considered patient-centred [22–25]. For the purposes of this review, we considered interventions that aimed to improve the patient-centredness of a TB service or programme as an end in itself as the gold standard. We also included interventions that aimed to improve the outcomes of people affected by TB (such as adherence and treatment outcomes) if the mechanisms for doing so was patient-centred (i.e., aimed at improving the experiences of people affected by TB). Authors had to explicitly refer to the intervention as 'patient-centred' or similar (see keywords outlined in section 2.2.). Interventions that aimed to improve patient outcomes without a patient-centred mechanism was not sufficient to be included in the review. We limited the search to between January 2005 and March 2020 (with

**Table 1. Types of projects included and excluded.**

| Included |
| --- |
| • Assessments of the needs of people affected by TB and intervention to address these needs in the TB service or programme. |
| • Patient-centred TB care interventions that aimed to improve the patient-centeredness of the TB service or programme. |
| • Evaluations of implemented patient-centred TB care enablers, such as psychosocial, material, or economic support. |
| • Mobile applications that aimed to facilitate and/or implement patient-centred care for TB, by, for example, providing information or psychosocial support. |

| Excluded |
| --- |
| • Conceptual descriptions/development of patient-centred care frameworks. |
| • Interventions in the TB service or programme aimed at improving adherence and/or treatment outcomes for people affected by TB without a patient-centred mechanism. This included interventions that incentivised adherence and treatment completion with financial or other rewards. |
| • Evaluations of the patient-centredness of a TB service or programme, or of adherence strategies such as directly observed therapy (DOT). |
| • Mobile applications to improve adherence and/or treatment outcomes in people affected by TB, without a patient-centred mechanism. |

the most recent search conducted on 23 March 2020) and to English-language manuscripts / reports. We selected this period because it is ten years preceding and five years after the WHO released guidance calling for patient-centred TB care and we sought to understand how programmes were aligned to the WHO call. We discuss our findings from this period relative to more recent publications in the discussion section.

We excluded conceptual research, evaluations of interventions that aimed to improve adherence or treatment outcomes without a patient-centred mechanism, and research that measured or evaluated how patient-centred a TB service or programme is. We also excluded interventions that incentivised adherence with financial or other rewards as we considered incentives contrary to the principles and aims of patient-centred care. Table 1 outlines the types of projects included and excluded.

## 2.2 Research articles: Process of identifying and screening records

We searched on PubMed and EBSCOhost databases using multiple keyword combinations related to patient-centred care and TB. Our keyword searches included 'tuberculosis' and variations of the following: patient/person/people/family -centred, -focused, -oriented, -centric. Further we conducted searches using 'tuberculosis' and either 'social support', 'economic support', 'socio-economic support', or 'enablers' (see S2 Appendix for the full electronic search strategy).

We exported the results (title and abstract) and removed duplicates. As a first step in the screening process, the first reviewer excluded records that were not related to TB, patient-centred care, or about enabling/supporting TB care and treatment. Two reviewers then separately reviewed the remaining results (title and abstract) against the inclusion/exclusion criteria, resolving any discrepancies in consultation with co-authors. Following this initial screening process two reviewers read and reviewed the full text of each remaining research article, resolving discrepancies between them, or, where an agreement could not be reached, consulting with a third reviewer.

## 2.3 Google search engine, website, TB conference, and reference list searches

We searched for grey literature on patient-centred TB care interventions using the Google search engine, TB-related websites such as non-government and globally funded TB

programme organisations, and the abstract booklets from the Union World Conference on Lung Health held annually between 2005 and 2020, using our keywords and inclusion and exclusion criteria. We limited the Google search engine results to the first 10 results pages.

We manually searched the reference lists of systematic reviews and best practice documents on patient-centred TB care that were identified through our database, Google, and website searches. We also searched the reference lists of the full text research articles included in the analysis. These searches served as a proxy measure for the comprehensiveness of the review method.

## 2.4 Follow-up discussions

Where possible, we contacted the authors to gain further insights into how they implemented their TB patient-centred care intervention(s).

## 2 5 Data analysis and synthesis

Analysis involved 'charting' the review results. We grouped the different patient-centred TB care interventions into thematic areas based primarily on intervention focus area(s) and outlined the details of each intervention, such as the setting, TB-type, target population, approach/methodology, and where along the TB care cascade [26] the intervention was focused. This process allowed us to identify patterns in how patient-centred TB care was implemented and to consider intervention strengths and limitations. We shared these initial findings with co-authors and expert others, enhancing our interpretations.

## 3. Results

Our searches yielded a total of 908 unique peer-reviewed research articles (Fig 1). In total, 13 research articles were included in a full text review, with 12 research articles identified through our database searches, and 1 through our manual reference list searches. Further, we identified 7 best practice documents and project reports through our grey literature searches on Google and other TB-related websites, and 19 conference abstracts.

## 3.1 Review of full-text articles

We found that patient-centred interventions were operationalised in the following ways: a) offering emotional/psychosocial support to people affected by TB (n = 6); (b) socio-economic relief for people affected by TB (n = 3); (c) targeted enhanced support (emotional, psychosocial, material etc.) for people affected by TB identified as at risk of becoming lost to follow-up (LTFU) (n = 2); (d) decentralising TB care to bring health workers closer to patients' everyday experiences (n = 1); and (e) stigma reduction interventions to remove, reduce, or mitigate negative patient experiences during care (n = 1) (S3 Appendix).

**3.1.1 Emotional/Psychosocial support interventions.** Six of the thirteen studies operationalised patient-centred TB care as emotional and psychosocial support to people affected by TB and/or their families [27–32]. Chalco et al. explored the forms and means of emotional support community nurses provided to people with drug-resistant TB (DR-TB) in Lima, Peru, using ethnographic research [27]. Working as part of a multi-disciplinary team, community nurses were found to provide valuable emotional support to patients both formally and informally that focused on problems related to different stages of treatment, TB stigma, adherence, side effects, socio-economic difficulties, death, and comorbidities. The outcomes of the emotional support were not measured among patients. Acha et al. described the impact of a psychosocial support group intervention for people with DR-TB and their families on treatment

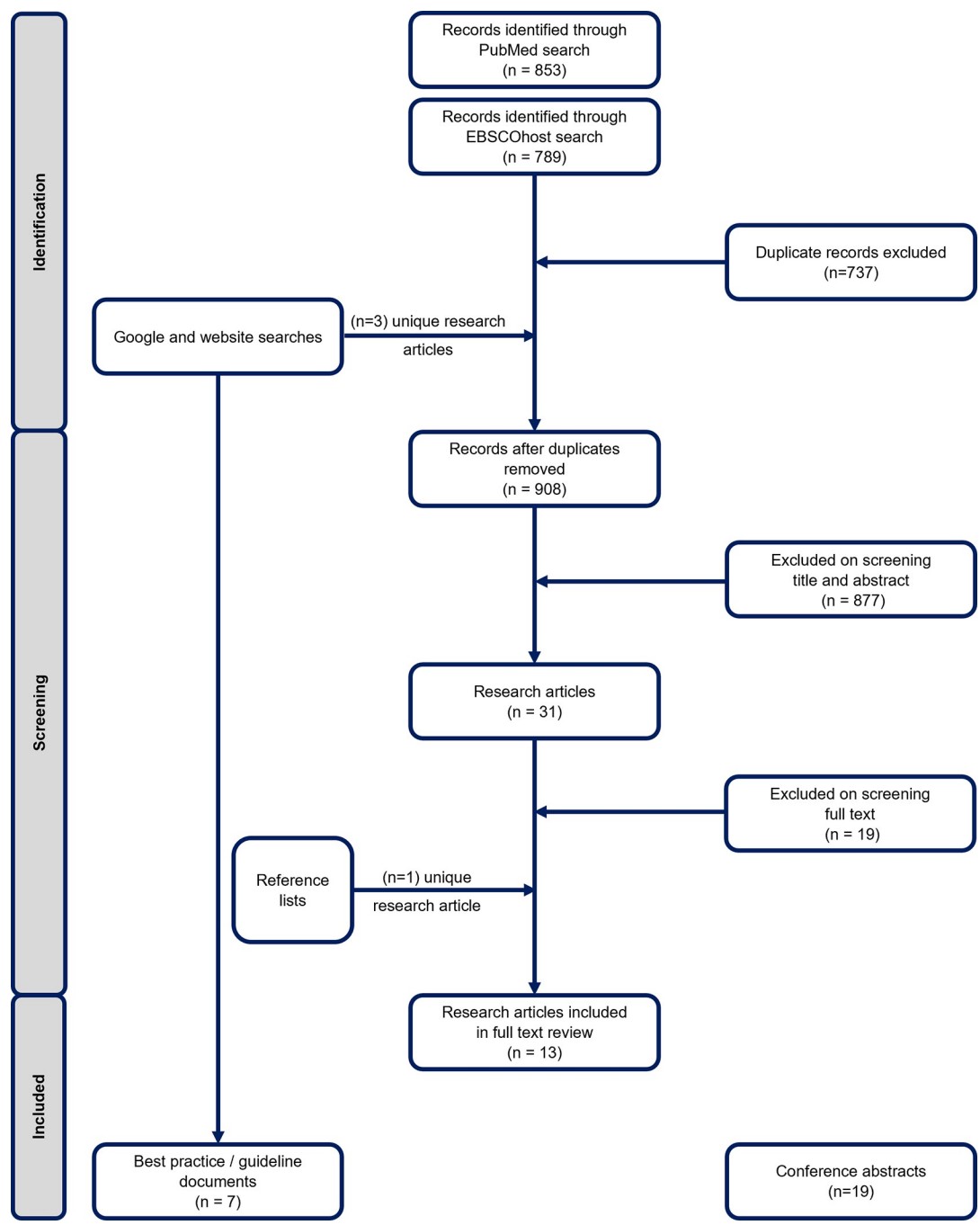

**Fig 1. Process and outcome of identification and screening of search results.**

outcomes and reported that the intervention normalised patients' experiences of TB disease and treatment [28]. After the intervention, excluding the 31.6% who remained in treatment, only 3.5% were LTFU. Adepoyibi et al. described a patient education and counselling intervention implemented in response to an outbreak of DR-TB in Papua New Guinea [32]. The intervention focused on providing emotional support and education to patients through a comprehensive and individualised package of counselling sessions delivered by peer counsellors, and with referral to a range of external services to address other clinical and social

challenges. Counsellors' efforts reduced LTFU among 331 people with DR-TB from 18% to 4% between the 2014 to 2015 cohorts. Khanal et al. and Walker et al. developed and measured the feasibility and acceptability of a psychosocial support package for people with DR-TB and their families in Nepal [29, 30]. The resulting intervention included education materials given to all patients and their family members, screening for depression, behavioural activation counselling as needed, and voluntary participation in support groups. Their mixed-methods evaluation showed that the intervention was acceptable to patients, and that feasibility was low for implementation into the National TB Programme. Li et al. implemented a community-based trial in Hubei Province, China, to explore elderly TB patients' experience of social support interventions [31]. Control group participants received a health education intervention alone, and those in the intervention group received psychotherapy and family and community support interventions in addition to health education. The social support level of patients was evaluated by the Social Support Rating Scale; comprehensive social support interventions increased the social support of elderly patients, compared to health education alone.

**3.1.2 Socio-economic relief for people affected by TB.** Baral et al.'s study evaluated the effectiveness of two strategies for social and economic support, i.e., counselling alone, or combined counselling and financial support, on treatment outcomes [33]. The cure rate for counselling alone was 85%, 76% for the combined counselling and financial support, and 67% for the group receiving no support. Fuady et al.'s study measured the potential for financial support to reduce the incidence of catastrophic costs for people affected by TB [34]. Supports included social protection for consultation and travel fees, food and drug costs, and income loss. The study found that cash transfers for a combination of income loss, travel costs, and food-supplement costs would have the greatest impact on the incidence of catastrophic costs but that they would not eliminate it altogether. Bhatt et al. retrospectively measured the impact of an integrated patient support package on the treatment outcomes of people affected by DR-TB which included cash handouts, transportation costs, nutritional supplements, psychosocial support, and ancillary medical aid [35]. Treatment outcomes were significantly better in the supported group (65% vs 46%) and death rates were reduced by more than 50%.

**3.1.3 Targeted enhanced support interventions for patients at risk of becoming LTFU.** Two studies implemented an enhanced support program for patients at risk of LTFU [36, 37]. Gelmanova et al. described the impact of an intense patient-centred treatment support intervention ('Sputnik') for people affected by TB (both drug-susceptible (DS)- and DR-TB) at risk of LTFU due to alcoholism, drug abuse, history of incarceration, and homelessness in Tomsk City, Russian Federation [36]. The intervention attempted to find programmatic solutions to social and economic barriers that prevented the successful completion of treatment. The study compared baseline characteristics, adherence before and during the intervention, and treatment outcomes, and found that adherence and treatment outcomes of both people with DR and DS-TB significantly improved during the program. Snyman et al.'s intervention involved enhanced adherence support to people with DR-TB at risk of LTFU by providing decentralised treatment and care [37]. Home visits provided the opportunity for patients to discuss reasons for treatment interruption with health workers and an opportunity to screen for mental health and substance use. Individualised management plans were developed by a multidisciplinary team following the home visits. Analysis of qualitative interviews and focus group discussions with providers and patients showed that patients valued the support provided.

**3.1.4 Decentralised TB care to connect health workers and patients.** Brust et al.'s study implemented an integrated home-based treatment model for DR-TB and HIV as patient-centred care [38]. Among the 67 (84%) patients who completed the intensive phase of treatment, all had converted their sputum culture to negative in a median of 55 days. The study

demonstrated feasibility of a home-based model of care for DR-TB treatment, and suggested that the involvement of family members, close monitoring of patients at home, and empowerment of patients by providing intensive treatment literacy were critical to improving treatment outcomes and may better serve patients' needs than conventional inpatient care.

**3.1.5 TB stigma reduction interventions.** Macq et al. aimed to reduce internalised stigma experienced by people affected by TB through an intervention that included TB self-help groups and 'patient-centred home visits' [39]. TB completion and cure rates and TB-related internalised stigma were measured. No significant difference in TB treatment outcomes was found. However, the intervention group showed a statistically significant reduction in internalised stigma scores after two months, compared to the control group.

## 3.2 Excluded research articles with notable contributions to how patient-centred TB care may be implemented

There were several studies excluded from a full text review as they did not meet our inclusion criteria, but which make notable contributions on how patient-centred TB care can be implemented.

Five studies framed patient-centred care as providing patients with the choice between facility-based DOT and home-based DOT with a treatment supporter of their choice [40–44]. These studies provided promising evidence for the benefits of allowing patients a degree of autonomy over how their treatment is administered. These studies illustrated how alternative treatment delivery methods could help overcome access and facility-level barriers for people affected by TB, and showed similar [40, 42, 43] and in some cases, improved [41], adherence and treatment outcomes than were achieved with facility-based DOT.

Five studies implemented socioeconomic support to people affected by TB during their treatment journey as patient-centred care, evaluating the relationship between such support and adherence and treatment outcomes [45–49]. Socioeconomic supports included financial support, transportation costs, and food parcels or vouchers, often delivered by way of incentive. These studies provided evidence of a positive impact of these supports on adherence, LTFU, and treatment outcomes. They also offered insight into operational challenges such as fidelity [47] and logistics and administration [45, 48], as well as considerations such as household size in determining appropriate financial support [48, 49].

Three studies focused on mobile applications to support treatment adherence of people affected by TB as patient-centred care [50–52]. The mechanisms for supporting treatment adherence using mHealth technologies included digital monitoring and self-reporting of adherence [50], Video Directly Observed Therapy [51], and SMS-based medication reminders [52]. Patient-participants in the three studies shared positive experiences of mHealth adherence support. Limited evidence on the impact of such applications on patients' experiences, treatment adherence and health outcomes is available however, and the scalability of mobile applications still needs to be evaluated.

## 3.3 Project reports and best practice documents

We identified 7 best practice documents related to patient-centred TB care (S4 Appendix). These documents were reviews of country-specific social support interventions similar to those described in the review of published articles [53–56] or offered patient-centred models and tools to guide and support implementation of patient-centred TB care interventions [15, 57]. These interventions reinforce the importance of embedding patient-centredness in TB services and offer useful tools for implementing patient-centred care principles. One of these, the TB Control Assistance Patient Centred Approach Strategy, introduced 5 focus areas to

realise patient-centred care in TB control programmes, with concomitant tools to guide implementation in each area [15, 57]. Key areas for intervention included engaging stakeholders, recognising patient rights, enabling partnerships between providers and patients, empowering and activating patients and communities, and monitoring and documenting activities and approaches.

### 3.4 Conference abstracts

We identified 19 conference abstracts to include in our review (S5 Appendix). Patient-centred TB care interventions presented in the abstracts focused on: emotional and psychosocial support (n = 5), decentralised DR-TB care (n = 5), enhanced support for high-risk patients (defined as at risk of insufficient adherence or of LTFU) (n = 4), socio-economic relief (n = 3), and training for health workers (n = 2). The findings from these abstracts supported the published articles reviewed, offering similar examples for implementing patient-centred TB care. Additionally, two abstracts shared findings from interventions with health workers to implement patient-centred TB care which were not reported in the published articles.

### 3.5 Follow-up discussions with researchers and project implementers

Follow-up discussions with authors of four of the interventions included in the review supported a more comprehensive understanding of the motivations and aims underlying specific interventions. Discussion also allowed insight into less tangible aspects of care–such as inequalities between providers and patients–that require intervention to enable patient-centred TB care.

Drawing on their experiences of implementing patient-centred TB care interventions in Khayelitsha, South Africa [37, 55], the authors emphasised that people affected by TB often faced barriers to accessing and completing treatment that are insurmountable without targeted intervention and support. The authors suggested that adherence be understood as fluctuating over the course of a person's life due to social and other circumstances. They argued that such an approach is needed to provide patients with the individualised support they desire.

Discussions with researchers in Papua New Guinea revealed that the guiding principle for the psychosocial support intervention was to balance power relations between patients and the health system [32]. They explained that people affected by TB experienced services as disempowering in traditionally hierarchical health systems. In the TB programme, historically aimed at TB surveillance and control, people affected by TB seldom felt that they were able to guide their own treatment journey. Implementers of a psychosocial support intervention for people with DR-TB in Peru reported the transformative power of providing emotional and psychosocial support to people affected by TB [28]. Through this intervention they were able to address health issues such as depression and social isolation, which are often neglected in health services, particularly among those with DR-TB.

## 4. Discussion

The findings of our review showed limited results on implementing patient-centred care for TB programmes, globally. Of the documents reviewed, patient-centred care was used primarily as a mechanism to improve adherence, reduce LTFU, and improve treatment outcomes. Interventions included components focused on patient and family education and counselling, and psychosocial and socioeconomic support (Fig 2). Further, these intervention components were predominantly focused on treatment initiation and adherence steps along the TB care cascade (Fig 3). The implementation of patient-centred TB care interventions to date have neglected to include crucial aspects of TB control such as prevention, reinfection or recurrence, or support

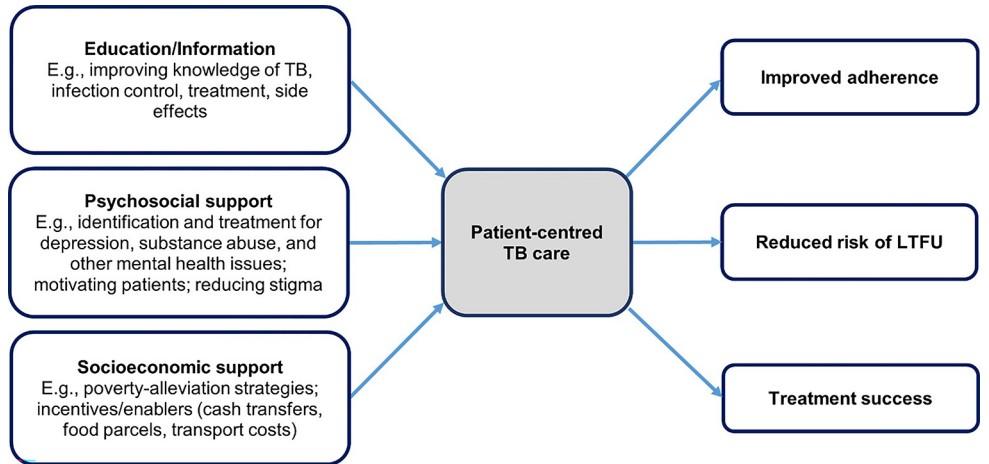

**Fig 2. Patient-centred TB care as an instrumental function.** The literature showed that patient-centred TB care is implemented as primarily having an instrumental function–aimed at improving adherence, lost to follow-up (LTFU), and treatment outcomes for people affected by TB. We found few examples of interventions that used patient-centred care indicators as measures of impact.

for patients re-entering the workforce. While these interventions showed promise in terms of supporting patient journeys by improving adherence and treatment outcomes, we found only three interventions that evaluated impact independently of adherence and treatment outcomes. Li et al. aimed to impact the level of social support experienced by elderly TB patients in Hubei Province, China, and measured the intervention's effectiveness using the Social Support Rating Scale [31]. Fuady et al.'s study aimed to measure the effect of financial support on the incidence of catastrophic costs for households affected by TB in Indonesia [34]. In Nicaragua, Macq et al. measured TB internalised social stigma and treatment outcome against an intervention that included TB clubs and patient-centred home visits [39].

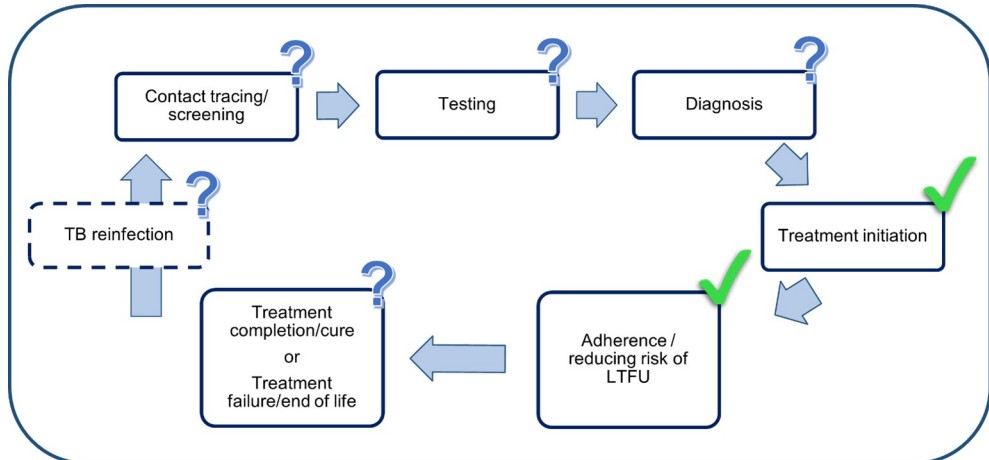

**Fig 3. Patient-centred TB care interventions as focused along the TB care continuum.** Most patient-centred TB care interventions were focused on the treatment initiation and adherence steps along the TB care cascade. Rather than be discretely applied to steps in the care cascade, patient-centred care should permeate all aspects of the care cascade–it is an approach to health care delivery which means that the needs and experiences of people affected by TB must be central to all aspects of care delivery [6, 15]. TB = tuberculosis. LTFU = lost to follow-up.

This is a notable gap in the intervention literature–not least for TB programmes, but for HIV as well and for patient-centred care in general [58, 59]. This gap is one that will arguably remain until we move beyond conventional conceptualisations of TB control. Arsenault, Roder-DeWan, and Kruk call attention to undermeasured indicators of TB care quality which include measures of the user experience, such as providers' attitudes, communication, as well as respect for privacy and confidentiality, and average patient wait times [60]. Such measures of care quality–which includes indicators of competent care and health systems–are arguably understood to be indirect measures of patient-centred care. They however omit systematic measurement of how we have seen patient-centred care operationalised in the review, as, for example, emotional and psychosocial support, socio-economic relief, decentralised care, and enhanced support. Nor do they measure, for example, the trust and confidence that patients have in their providers and service, or to what extent patients feel supported, empowered, and informed to manage their disease and treatment, as outlined in definitions of patient-centred TB care [16, 18]. Several studies and projects had features of patient-centred care as guiding principles or implicit motivations underlying their intervention(s). However, these principles were similarly not systematically operationalised and measured across the TB care cascade as promoted in international guidelines [61]. Additionally, most of the interventions reviewed focused on people affected by DR-TB and on identifying patients at risk of LTFU, with few interventions focused on DS-TB, or on the TB programme more broadly. This omission may be due to the limited availability of resources in public health systems, resulting in the need to prioritise interventions that reach patients with more severe TB disease (such as those with DR-TB) and those most at risk of poor health outcomes. It may also be indicative of approaches in health services (e.g., quality improvement processes) for effecting improvements with targeted inputs rather than considering fundamental changes to the way in which services are imagined and delivered [23].

USAID's TB Control Assistance Programme provides tools for measuring the quality of TB care from the patient perspective, covering both the performance and importance of nine TB care dimensions for patients [15, 62]; eight of these quality dimensions have been tested and validated through statistical analysis. These tools were piloted in five countries as part of a patient-centred care approach and could inform context specific interventions and benchmarks for improving TB services in these settings [57]. They include focus group discussions with people affected by TB to rank quality dimensions based on their relative importance to patients and questionnaires to assess health facility performance. While the tools do not offer patient-centred care indicators to measure for each individual patient similar to indicators for adherence, LTFU, and treatment outcomes, they provide a foundation from which to consider how to do so routinely. This includes measures of communication and information, affordability, support, stigma, and patient-provider relationship, amongst others.

Another notable limitation in how patient-centred care for TB has been implemented, includes an overt focus on TB as primary disease, to the exclusion of other health conditions (e.g., antenatal and mother and child health) and comorbidities (most notably HIV, diabetes, and hypertension). Further, children and adolescents were largely excluded in approaches to delivering patient-centred TB care (only one intervention included consideration of patient-centred care for adolescents [32]), and interventions did not distinguish between how men and women may be differently affected by TB or challenged to engage in TB treatment. These groups are highlighted in the Stop TB Partnership's recent Community, Rights, and Gender Tuberculosis Assessment, along with calls for a better understanding of community, legal, and gender contexts in TB responses [63].The Global Plan to End TB 2023–2030 similarly emphasises integrated service delivery that includes management of comorbidities as noted above, and brings attention to the specific needs of children, adolescents, men and women, and other

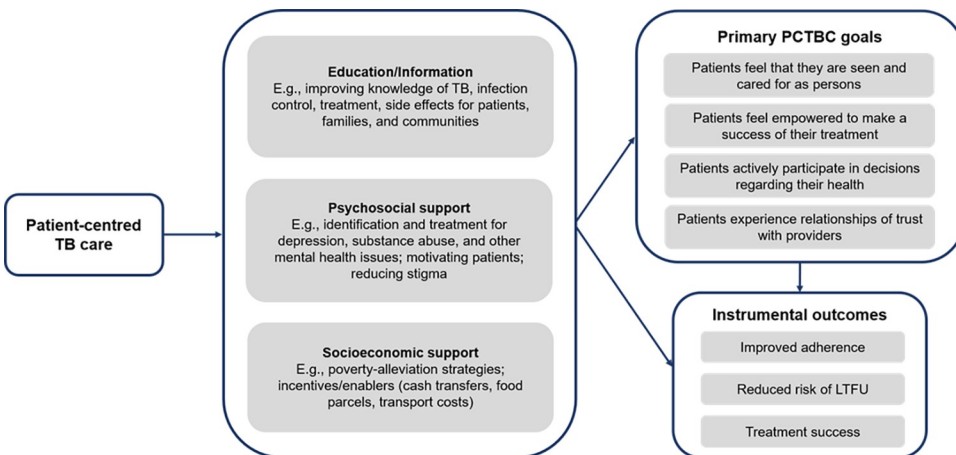

**Fig 4. Pathways to achieve primary and instrumental outcomes with operationalising patient-centred TB care holistically.** Patient-centred interventions for TB can be attached to both instrumental treatment-related TB outcomes and patient-centred outcomes as illustrated. The key message is that interventions that aim to improve the experiences of people affected by TB by acting on the TB education and information available to them, or their psychosocial and socioeconomic support, should lend themselves to be measured against PCTBC outcomes. PCTBC = patient-centred TB care. LTFU = lost to follow-up.

key and vulnerable populations [18]. The document propounds 'people-centred care' as a priority action to scale up TB diagnostics and care, and signals that implementing such care requires TB care to be reimagined.

We similarly argue that fundamental change and reimagining is needed in TB services (and in health in low and middle-income settings more generally), for patients to be engaged as partners in care. Such a reimagining must include more information, alternative and better treatment options, more counselling, and material support where there are significant socioeconomic barriers [6, 64]. Furthermore, such reimagining requires work with health workers to shift attitudes to recognise patients as partners, and to nurture relationships of trust between people affected by TB and the health service [15, 16]; how frontline health workers internalise and support patient-centred care principles ultimately influences how guidelines and policies become practice. These principles should be considered an essential part of every step along the TB care cascade. Reinforcing and implementing improvements in the quality of the TB service will have a positive impact on programme performance [6]. However, delivering genuinely patient-centred TB care requires more than programmatic improvements: it requires shifting the programme away from one that has historically been centred on disease control towards one that recognises the needs of people affected by TB, aims to empower them, and to foster relationships of trust between patients and providers [16, 18].

While a reimagining of TB services may be the ideal, we suggest that there are three key areas for implementing patient-centred TB care in the immediate timeframe: (1) education/information; (2) psychosocial support; and (3) socio-economic support (Fig 4). Addressing these key areas must extend beyond the treatment period to facilitate holistic social and economic recovery and reintegration of people affected by TB into their communities. Patient-centred care interventions must be sensitive to people's gender, age, socio-economic status, and health concerns and conditions. We cannot emphasise enough that when patient-centred TB care interventions are designed and implemented, they must measure the impact on the patient experience alongside treatment and health outcomes, as only three of the studies in this review have done [31, 34, 39].

This review has several strengths. First, the sources reviewed were comprehensive and included the reference lists of systematic reviews, full text articles, and grey literature in the form of policy, best practice, and project report documents. Second, we used a broad set of keywords to conduct searches of research databases, without limiting to specific settings. Third, our process for conducting the review included four reviewers. Fourth, our inclusion and exclusion criteria were developed and iteratively refined in discussion with co-authors during the screening process, which allowed for a targeted focus on finding examples of how patient-centred care for TB has been implemented. Fifth, follow-up discussions with researchers are a strength of scoping review methodology [20], allowing further insight into findings. The review was limited to publications until March 2020, and articles published in English. The review also only considered TB-focused interventions and not patient-centred interventions for health and well-being.

## 5. Conclusion

Patient-centred care has become a noteworthy principle to strive for and implement in health systems worldwide. The role of the patient or person has risen to prominence in various treatment models (such as those for HIV), with the recognition that a patient who is informed, engaged in care, and empowered to manage their treatment is an asset to the health system and promotes positive programme outcomes. In the collective TB world, this concept is, however, still novel. Our review has shown that while there is growing interest in the development of patient-centred TB care interventions, up to 2020, few interventions had been implemented, and interventions often face serious shortcomings in terms of engaging patients and communities in the planning of services at every step of the TB care cascade. While we have outlined recommendations for improving patient-centred TB care, further research is needed to explore how to prioritise patient-centred care in resource constrained contexts. Finally, if it is to be truly realised then health systems must incorporate reliable measures of patient-centred care outcomes in routine health services.

## Supporting information

**S1 Appendix. PRISMA-ScR extension checklist.**
(PDF)

**S2 Appendix. Full electronic search strategy.**
(DOCX)

**S3 Appendix. Full text research articles included in the review, grouped by implementation of patient-centred TB care.** DR-TB = drug-resistant TB. SAT = self-administered therapy. XDR-TB = extensively drug-resistant TB. LTFU = lost to follow-up. DOT = directly observed therapy.
(DOCX)

**S4 Appendix. Project reports and best practice documents included in the review.**
DR-TB = drug-resistant TB.
(DOCX)

**S5 Appendix. Conference abstracts included in the review, chronologically organised.**
DOT = Directly Observed Therapy. DR-TB = drug-resistant tuberculosis. LTFU = lost to follow-up. ART = antiretroviral therapy.
(DOCX)

## Author Contributions

**Conceptualization:** Hanlie Myburgh, Dzunisani Baloyi, Marian Loveday, Graeme Hoddinott.

**Formal analysis:** Hanlie Myburgh, Dzunisani Baloyi, Marian Loveday, Graeme Hoddinott.

**Methodology:** Hanlie Myburgh, Dzunisani Baloyi, Marian Loveday, Sue-Ann Meehan, Muhammad Osman, Dillon Wademan, Graeme Hoddinott.

**Project administration:** Hanlie Myburgh, Marian Loveday.

**Supervision:** Anneke Hesseling, Graeme Hoddinott.

**Validation:** Hanlie Myburgh, Marian Loveday, Sue-Ann Meehan, Muhammad Osman, Dillon Wademan, Anneke Hesseling.

**Visualization:** Hanlie Myburgh.

**Writing – original draft:** Hanlie Myburgh.

**Writing – review & editing:** Hanlie Myburgh, Dzunisani Baloyi, Marian Loveday, Sue-Ann Meehan, Muhammad Osman, Dillon Wademan, Anneke Hesseling, Graeme Hoddinott.

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
