## [Decision Letter · Decision Letter 0]

9 Sep 2022

PGPH-D-22-01185

A scoping review of patient-centred tuberculosis care interventions: gaps and opportunities

Dear Dr. Myburgh,

Thank you for submitting your manuscript to PLOS Global Public Health. After careful consideration, we feel that it has merit but does not fully meet PLOS Global Public Health’s publication criteria as it currently stands. Therefore, we invite you to submit a revised version of the manuscript that addresses the points raised during the review process.

To be considered for publication, please address all the reviewer feedback, but in particular, the methodological concerns raised by reviewer #1 - inclusion / exclusion criteria and compliance with PRISMA guidelines.      

We look forward to receiving your revised manuscript.

Kind regards,

Associate Professor Suman Majumdar

Academic Editor

Journal Requirements:

If you did not receive any funding for this study, please simply state: “The authors received no specific funding for this work.

Additional Editor Comments (if provided):

Reviewers' comments:

Reviewer's Responses to Questions

**Comments to the Author**

1. Does this manuscript meet PLOS Global Public Health’s publication criteria? Is the manuscript technically sound, and do the data support the conclusions? The manuscript must describe methodologically and ethically rigorous research with conclusions that are appropriately drawn based on the data presented.

Reviewer #1: Partly

Reviewer #2: Yes

Reviewer #3: Yes

2. Has the statistical analysis been performed appropriately and rigorously?

Reviewer #1: N/A

Reviewer #2: Yes

Reviewer #3: Yes

3. Have the authors made all data underlying the findings in their manuscript fully available (please refer to the Data Availability Statement at the start of the manuscript PDF file)?

Reviewer #1: Yes

Reviewer #2: Yes

Reviewer #3: Yes

4. Is the manuscript presented in an intelligible fashion and written in standard English?

Reviewer #1: Yes

Reviewer #2: Yes

Reviewer #3: Yes

5. Review Comments to the Author

Reviewer #1: This paper aimed to conduct a scoping review of patient-centered TB interventions, which would be an important advance in the literature, given the importance of and increased focus (e.g., in global guidelines) on patient-centered TB care.

Many of my comments/questions related to tensions between the inclusion and exclusion criteria listed in Table 1:

• Table 1/lines 113-116: It is not clear what the rationale is for excluding “evaluations of interventions that aimed to improve adherence or treatment outcomes”. What if these interventions aimed to improve adherence or treatment outcomes by improving patient-centeredness? Although having interventions that respect individuals and focus on their needs is important in and off itself, isn’t improving outcomes an additional important potential benefit of more patient-centered care? It would be useful for the authors to clarify the rationale for this exclusion. It would also be useful to better explain how it interacts/overlaps with the inclusion criterion “Evaluations of implemented patient-centred TB care enablers, such as psychosocial, material, or economic support”, as there were several instances throughout the paper in which it was not clear why certain articles were included or excluded and this likely relates to tension between the exclusion and inclusion criteria – such as:

o The first few paragraphs of the results section include summaries of how interventions from several of the articles reduced LTFU – how was this not violating the exclusion criterion?

o It is not clear why relevant systematic review(s) are not discussed – such as the 2016 PLOS One article by van Hoorn et al (The Effects of Psycho-Emotional and Socio-Economic Support for Tuberculosis Patients on Treatment Adherence and Treatment Outcomes – A Systematic Review and Meta-Analysis). This review would seem to also evaluate “patient-centred TB care enablers, such as psychosocial, material, or economic support”?

o Results section 3.2 on excluded articles: it is not clear why these articles were excluded. For example, several studies in this section are related to economic support – why does this not fit into exclusion criterion 3? Perhaps it could be helpful to explain why these articles were excluded in this section.

o Lines 270-271: “Three studies focused on mobile applications to support treatment adherence of people affected by TB (45–47).” Inclusion criteria #4 is “Mobile applications that aimed to facilitate and/or implement patient-centred care for TB, by, for example, providing information or psychosocial support” – so why were these articles excluded?

Other comments/questions:

• In order for PRISMA scoping review guideline 8 to be met (“Present the full electronic search strategy for at least 1 database, including any limits used, such that it could be repeated”) I believe the exact search terms need to be provided (i.e., in an appendix). The current description on page 6 doesn’t provide enough information for the search to be replicated. For example, if I do a PubMed search with the terms “tuberculosis AND ("social" OR "socio-economic" OR "economic") AND support” limited to English-language articles between Jan 1 2005 and March 23 2020, 2532 articles are returned, which is far more than the 853 articles mentioned in Figure 1.

• Lines 280-283: “These interventions reinforce the importance of embedding patient centredness in TB services and offer useful tools for implementing patient-centred care principles. However, they do not introduce novel evaluations of interventions which have not already been described in the review of published articles.” The introduction and methods leave the impression that the purpose of the article is not just to evaluate patient-centered interventions (indeed, this seems to fit with the broader purposes of a scoping review and justifies the authors decision not to conduct a systematic review). But in this and some other places in the results, it seems as though only papers that reported some sort of impact/outcome are discussed. Were papers that did not include impact on adherence/treatment outcomes/some other outcome explicitly excluded? If so, this should be clarified in the methods. If not, it would still seem relevant to this review to discuss the interventions discussed in these project reports and best practice documents, even if the impact of these interventions was not evaluated.

• Section 3.4 – it would be useful to categorize the interventions found in the conference abstracts using the same categorizations that were used for the published studies (in Section 3.1) or the categorizations used in Figure 4. It would also be useful to add these categorizations to the appendix tables that summarize the articles.

• A point emphasized in the discussion is the need to track “patient-centred care outcomes” rather than just adherence/LTFU/treatment outcomes. Some examples of articles that do this are mentioned, but very few details are provided on what the outcomes were that were measured. Given the dearth of studies measuring these outcomes, most readers of this paper may not be very familiar with the concept of patient-centred care outcomes. The authors should consider discussing these outcomes in more detail (e.g. for example in the results/through a table or appendix table) – potentially including not just those that have been evaluated in studies but also those that were proposed (for example in the Arsenault, Order-DeWan, and Kruk article). Is there a reason why these outcomes are not included in Figure 4?

Reviewer #2: The focus on "patient" or "people" centered care is an important area of analysis. The existing publication may be further strengthened with more in-depth consideration of human rights and gender related barriers to accessing quality TB services. Noting that a TB CRG Assessment has been completed in South Africa and that a global level analysis of human rights and gender related barriers has also been published in a peer reviewed context as well (https://www.hhrjournal.org/2021/12/building-the-evidence-for-a-rights-based-people-centered-gender-transformative-tuberculosis-response-an-analysis-of-the-stop-tb-partnership-community-rights-and-gender-tuberculosis-assessment/ ) . Finally it may be useful to also look at the recently released Global Plan to End TB 2023-2030 and draw further on the practical elements of person centered care.

Reviewer #3: The submission of the manuscript is timely and critical in view of the operational lack of emphasis on what is a critical pillar in the end TB strategy. The period included of 2005 to 2020 is relevant particularly since it included the recognition of the compounded problems of an ever increasing situation of DR TB.

The methods and inclusion criteria are sound and the definitions and findings articulated well.

Perhaps the search needed to include policy documents of large TB control programmes like the NTEP to demonstrate which issues of patient centric care are highlighted there and triangulate these with implementation studies in that context. The contacting of authors for implementation insights is also commendable.

One of the seminal studies on PCC from Peru was the Lima study of Paul Farmer on DR TB patients. This and the Papworth study in the UK could have been mentioned as a background or at least in the Discussion.

The manuscript may like to include which of the interventions in PCC have been actually incorporated in policy.

Line 267 mentions about the role of financial support. Maybe good here to expand a little citing the reasons why it alone is not sufficient for offsetting catastrophic costs.

The authors may also like to provide emphatic guidance to readers as to which approach for PCC yields the most optimal patient outcomes and the authors have repeatedly stressed the importance of measuring the outcomes.

Studies on delivery of integrated comprehensive care in adults and children is another important point mentioned by the authors which needs mention of the specific studies that were undertaken for arrival at a conclusion.

Similarly has a comparative analysis been done on the effect of virtual/remote or in person support. Studies on patient preferences maybe highlighted.

The review should also question whether the concept of patient oriented care (PCC) has been ingrained in the front line workers in the various countries as a part of their in service or pre service training.

6. PLOS authors have the option to publish the peer review history of their article (what does this mean?). If published, this will include your full peer review and any attached files.

**Do you want your identity to be public for this peer review?** For information about this choice, including consent withdrawal, please see our Privacy Policy.

Reviewer #1: **Yes: **Theresa Ryckman

Reviewer #2: No

Reviewer #3: No

---

## [Decision Letter · Decision Letter 1]

21 Dec 2022

A scoping review of patient-centred tuberculosis care interventions: gaps and opportunities

PGPH-D-22-01185R1

Dear Mrs Myburgh,

We are pleased to inform you that your manuscript 'A scoping review of patient-centred tuberculosis care interventions: gaps and opportunities' has been provisionally accepted for publication in PLOS Global Public Health.

Best regards,

Associate Professor Suman Majumdar

Academic Editor

Reviewer Comments (if any, and for reference):

Reviewer's Responses to Questions

**Comments to the Author**

1. If the authors have adequately addressed your comments raised in a previous round of review and you feel that this manuscript is now acceptable for publication, you may indicate that here to bypass the “Comments to the Author” section, enter your conflict of interest statement in the “Confidential to Editor” section, and submit your "Accept" recommendation.

Reviewer #1: All comments have been addressed

Reviewer #3: All comments have been addressed

2. Does this manuscript meet PLOS Global Public Health’s publication criteria? Is the manuscript technically sound, and do the data support the conclusions? The manuscript must describe methodologically and ethically rigorous research with conclusions that are appropriately drawn based on the data presented.

Reviewer #1: Yes

Reviewer #3: Yes

3. Has the statistical analysis been performed appropriately and rigorously?

Reviewer #1: N/A

Reviewer #3: N/A

4. Have the authors made all data underlying the findings in their manuscript fully available (please refer to the Data Availability Statement at the start of the manuscript PDF file)?

Reviewer #1: Yes

Reviewer #3: No

5. Is the manuscript presented in an intelligible fashion and written in standard English?

Reviewer #1: Yes

Reviewer #3: Yes

6. Review Comments to the Author

Reviewer #1: The authors have responded to my comments and suggestions. The scope of the review is now much clearer, and I think this will be a valuable addition to the literature.

Reviewer #3: Thanks for your response. Herewith below please find the references wrt Dr. Farmer's workin Lima and the Papworth experiment in the UK. You may include them in the manuscript at the appropriate places.

Mitnick C, Bayona J, Palacios E, Shin S, Furin J, Alcántara F, Sánchez E, Sarria M, Becerra M, Fawzi MC, Kapiga S, Neuberg D, Maguire JH, Kim JY, Farmer P. Community-based therapy for multidrug-resistant tuberculosis in Lima, Peru. N Engl J Med. 2003 Jan 9;348(2):119-28. doi: 10.1056/NEJMoa022928. PMID: 12519922.

Shin S, Furin J, Bayona J, Mate K, Kim JY, Farmer P. Community-based treatment of multidrug-resistant tuberculosis in Lima, Peru: 7 years of experience. Soc Sci Med. 2004 Oct;59(7):1529-39. doi: 10.1016/j.socscimed.2004.01.027. PMID: 15246180.

Bhargava A, Pai M, Bhargava M, Marais BJ, Menzies D. Can social interventions prevent tuberculosis?: the Papworth experiment (1918-1943) revisited. Am J Respir Crit Care Med. 2012 Sep 1;186(5):442-9. doi: 10.1164/rccm.201201-0023OC. Epub 2012 Jul 5. PMID: 22773730.

7. PLOS authors have the option to publish the peer review history of their article (what does this mean?). If published, this will include your full peer review and any attached files.

**Do you want your identity to be public for this peer review?** For information about this choice, including consent withdrawal, please see our Privacy Policy.

Reviewer #1: **Yes: **Theresa Ryckman

Reviewer #3: No
